# Evaluation of Sex Differences in Murine Diabetic Ketoacidosis and Neutropenic Models of Invasive Mucormycosis

**DOI:** 10.3390/jof7040313

**Published:** 2021-04-18

**Authors:** Teclegiorgis Gebremariam, Sondus Alkhazraji, Abdullah Alqarihi, Nathan P. Wiederhold, Laura K. Najvar, Thomas F. Patterson, Scott G. Filler, Ashraf S. Ibrahim

**Affiliations:** 1The Division of Infectious Diseases, The Lundquist Institute for Biomedical Innovation at Harbor-University of California at Los Angeles (UCLA) Medical Center, Torrance, CA 90502, USA; tghbremariam@lundquist.org (T.G.); salkhazraji@lundquist.org (S.A.); aalqarihi@lundquist.org (A.A.); sfiller@lundquist.org (S.G.F.); 2University of Texas Health Science Center at San Antonio, San Antonio, TX 78229, USA; wiederholdn@uthscsa.edu (N.P.W.); NAJVAR@uthscsa.edu (L.K.N.); patterson@uthscsa.edu (T.F.P.); 3South Texas Veterans Health Care System, San Antonio, TX 78229, USA; 4David Geffen School of Medicine at UCLA, Los Angeles, CA 90095, USA

**Keywords:** *Rhizopus*, *Mucor*, mucormycosis, murine, sex

## Abstract

There is increased concern that the quality, generalizability and reproducibility of biomedical research can be influenced by the sex of animals used. We studied the differences between male and female mice in response to invasive pulmonary mucormycosis including susceptibility to infection, host immune reaction and responses to antifungal therapy. We used diabetic ketoacidotic (DKA) or neutropenic mice infected with either *Rhizopus delemar* or *Mucor circinelloides*. The only difference detected was that when DKA mice were infected with *M. circinelloides*, female mice were more resistant to infection than male mice (median survival time of 5 vs. 2 days for female and male mice, respectively). However, a 100% lethality was detected among infected animals of both sexes. Treatment with either liposomal amphotericin B (L-AMB) or posaconazole (POSA) protected mice from infection and eliminated the difference seen between infected but untreated female and male mice. Treatment with L-AMB consistently outperformed POSA in prolonging survival and reducing tissue fungal burden of DKA and neutropenic mice infected with *R. delemar* or *M. circinelloides*, in both mouse sexes. While little difference was detected in cytokine levels among both sexes, mucormycosis infection in the DKA mouse model induced more inflammatory cytokines/chemokines involved in neutrophil (CXCL1) and macrophage (CXCL2) recruitment vs. uninfected mice. As expected, this inflammatory response was reduced in the neutropenic mouse model. Our studies show that there are few differences between female and male DKA or neutropenic mice infected with mucormycosis with no effect on the outcome of treatment or host immune response.

## 1. Introduction

There is a growing concern in the scientific community and lay public about the reproducibility of biomedical research. Study design elements that are critical to the reproducibility of results include, but are not limited to blinding, randomization and sample-size calculations [1,2]. Pre-clinical research with animal models may be difficult to reproduce, due to differences in the strain of animals that is used, laboratories and laboratory environments and subtle changes in protocols that may not be effectively communicated in publications [1,2]. In addition, there is increased recognition that the quality, generalizability and reproducibility of biomedical research can be influenced by sex of the experimental animal, and that the over-reliance on a single sex of animals in pre-clinical research may obscure key sex differences that could guide clinical trials [1]. Thus, the National Institutes of Health (NIH) now recognizes that the failure to account for sex (a biological variable defined by characteristics encoded in DNA, such as reproductive organs and other physiological and functional characteristics) as a key biological variable can undermine the scientific rigor, transparency and generalizability of research findings [3]. Because of this, applications to be considered for funding in fiscal year 2017 and beyond must report plans to balance male and female cells and animals in preclinical studies unless unwarranted based on rigorously defined exceptions, and that the absence of previous data in an area does not by itself constitute a strong justification to use only one sex [3].

Mucormycosis is a life-threatening, fungal infection caused by various species of the order Mucorales. While mucormycosis primarily occurs in patients immunocompromised by diabetic ketoacidosis (DKA), cytotoxic chemotherapy, immunosuppressive therapy and/or hematologic malignancies, immunocompetent patients with severe trauma are also at increased risk of contracting the infection [4,5,6]. While current treatment options for mucormycosis include the reversal of underlying predisposing factor, surgical debridement of infected foci and antifungal therapy, overall mortality rates remain between 40–90% [6,7,8]. Thus, the development of new therapeutic strategies for mucormycosis is of paramount importance. Owing to the rarity of the disease, it is difficult to conduct comparative clinical trials to evaluate treatment options for mucormycosis and there is a heavy reliance on animal models for evaluation and development of novel therapeutic strategies. Two of the murine models that have been developed and made available for the scientific community through the NIH/NIAID pre-clinical resources are models in which DKA is induced in mice or they are rendered neutropenic [9]. These two models represent two patient categories at high risk of mucormycosis [4,5,6]. We and others have used these models extensively to evaluate the efficacy of current and investigational drugs in treating mucormycosis and to understand the pathogenesis of infection mainly by using male mice [9,10,11,12,13,14,15].

Given the lack of knowledge on the effect of mouse sex on mucormycosis, we used the DKA and neutropenic mouse models to investigate differences between males and females in susceptibility to invasive pulmonary mucormycosis, immune responses to infection and antifungal therapy outcome using *Rhizopus delemar* and *Mucor circinelloides*, which are two of the most common causes of mucormycosis [7,16].

## 2. Materials and Methods

### 2.1. Organisms and Culture Conditions

*Rhizopus arrhizus* var. *delemar* 99–880 and *M. circinelloides* f. *jenssenii* DI15-131 are clinical isolates obtained from the Fungus Testing Laboratory at The University of Texas Health Science Center at San Antonio (UTHSCSA). These isolates had been utilized by our laboratory in numerous studies and resulted in consistent infections in animal models [9,17]. Both isolates were propagated on potato dextrose agar (PDA) plates for 4–6 days at 37 °C. Sporangiospores were collected in sterile phosphate buffered saline (PBS) containing 0.01% Tween 80, followed by two washes in PBS and the number of spores determined by hemocytometer. The number of sporangiospores per milliliter were adjusted for a working inoculum of 10^7^ spores/mL of PBS for *R. delemar* or 10^8^ spores/mL of PBS for *M. circinelloides*. Sporangiospore viability was determined by quantification of colony-forming units on PDA plates supplemented with 0.1% Triton X-100.

### 2.2. Antifungal Agents

Posaconazole (POSA) (Merck & Co., Inc., Rahway, NJ, USA) was purchased as an oral suspension (200 mg/5 mL) and kept at room temperature. Liposomal amphotericin B (L-AMB) (Gilead Science, San Dimas, CA, USA) was initially dissolved in irrigation water and diluted into 5% dextrose water per the manufacturer’s instructions. Both drugs were the clinical formulations and were obtained from a local pharmacy. All drugs were prepared fresh dosed per gram mouse body weight. 

### 2.3. Mouse Strains and Immunosuppression

Male or female ICR mice weighing ~25 g (Envigo, Indianapolis, IN, USA) were used in this study. Mice were housed 5 animals per cage with males and females in separate cages. Mice had access to food and water ad libitum throughout the course of the evaluations. To induce diabetes and slight ketoacidosis, freshly prepared streptozotocin in ice-cold citrate buffer (pH 4.2), was filter sterilized and immediately administered to mice by intraperitoneal injection at 210 mg/kg. Seven days after streptozotocin injection, glycosuria and ketonuria were determined by the use of keto-Diastix reagent strips (Bayer, Elkhart, IN, USA). Cortisone acetate (Sigma-Aldrich, ST. Louis, MO, USA) dissolved in 0.05% Tween 80 (Sigma-Aldrich) was also administered to mice (250 mg/kg) subcutaneously on Day −2 and +3, relative to infection [9]. 

Neutropenia was induced by administering cyclophosphamide and cortisone acetate. Cyclophosphamide (25 mg/mL; pharmaceutical grade) was dissolved in irrigation water and administered intraperitoneally at a dose of 200 mg/kg on days −2, +3 and +8, relative to infection. Cortisone acetate powder was freshly prepared as a suspension of 25 mg/mL in sterile physiologic phosphate buffered saline and 0.05% Tween 80. The suspension was administered subcutaneously at a dose of 500 mg/kg on days −2, +3 and +8, relative to infection. To prevent bacterial super-infection and deaths in the immunosuppressed mice, mice were given antibacterial prophylaxis consisting of enrofloxacin (enrofloxacin, Bayer, Leverkusen, Germany) at 50 ppm in the mice’s drinking water three days prior to infecting with fungal spores, then switched to subcutaneous ceftazidime (5 mg/mouse) on day 0 through day +8 for the DKA mice and through day +13 for the neutropenic mice [9]. 

### 2.4. Infection and Treatment

Mice were infected intratracheally with the appropriate spore inocula. First, mice were sedated by intraperitoneal injection of 0.2 mL of a mixture of ketamine 82.5 mg/kg (Phoenix, St. Joseph, MO, USA; prepared from a stock solution of 100 mg/mL of PBS) and xylazine 6 mg/Kg (Lloyd Laboratories, Shenandoah, IA, USA; prepared from a stock solution of 100 mg/mL of PBS) [9]. This dose delivered full anesthesia to the mouse for ~15–30 min. The mice were placed on their backs on heat pads that had been prewarmed to 37 °C and under heating lamps (at arm’s length) to prevent hypothermia. While pulling the tongue anteriorly and to the side with forceps, 25 µL of 2.5 × 10^5^ spores of *R. delemar* or 2.5 × 10^6^ spores of *M. circinelloides* was injected through the vocal cords into the trachea with a Fisher brand Gel-loading tip (Cat # 02−707-138) [9]. The mice were next placed on their backs on the heat pads until they recovered from the anesthesia. To determine the delivered fungal inoculum to the lungs, immediately after infection and prior to recovering from anesthesia, 5 mice per sex were randomly sacrificed, their lungs harvested, homogenized and quantitatively cultured on PDA plates containing 0.1% Triton X-100 and allowed to incubate at 37 °C for 48 h. 

Following infection, mice were randomly distributed into treatment groups. Uninfected mice that intratracheally received 25 μL of PBS alone. Mice were treated with either 15 mg/kg L-AMB (once daily [qd], administered through the tail vein) or 30 mg/kg POSA (twice daily [bid], given by oral gavage). Treatment was started 24 h post infection and continued for 4 days for L-AMB and 7 days for POSA. Infected untreated mice (placebo) received 5% dextrose water. The primary endpoint for efficacy was time to morbidity of infected mice through day 21 with moribund mice humanely euthanized. Secondary endpoints included assessment of fungal burden in lungs and tissues (primary and secondary target organs [9]) using conidial equivalents (CE)/g of tissue by quantitative PCR (qPCR) [18], and cytokine analysis of lungs homogenates or whole blood sera using a mouse magnetic Luminex assay from R&D systems (Cat # LXSAMSM-11) and conducted on mice sacrificed 4 days post infection. The cytokines that were measured were involved in regulating the innate immune response, and many had been reported to influence the host defense against infection caused by Mucorales [19,20,21]. These included: CCL2/MCP-1/JE, CCL3/MIP-1α, CCL4/MIP-1β, CXCL1/KC, IFN-γ, IL-1α, IL-1β, IL-17A, IL-23 p19, TNF-α and VEGF. The tissue fungal burden and cytokine analysis were conducted on the same mice to better correlate clearance mechanisms with the inflammatory immune response. All experiments were conducted in duplicate with at least 10 mice per group in each experiment with the exception of uninfected controls which had 3–5 mice per group per experiment. 

## 3. Results

### 3.1. DKA Mice Infected with R. delemar

In two independent experiments with similar results, we did not detect any significant difference in survival of male or female DKA mice infected with *R. delemar* 99–880 (5 and 6 days median survival time and 5% and 0% overall survival for female and male mice, respectively [*p* = 1.0]). Consistent with our previous published data in which we used male DKA mice [9], POSA and L-AMB enhanced median survival time and prolonged overall survival with L-AMB outperforming POSA. Specifically, POSA resulted in 10 day median survival time for both female and male mice, while L-AMB resulted in 18 days and >21 days median survival time for female and male mice, respectively. Further, POSA and L-AMB treatment caused ~30% and ~50% overall survival (Figure 1A). Importantly, both POSA and L-AMB showed similar efficacy in both female and male DKA mice (*p* = 0.571 and *p* = 0.431 for POSA and L-AMB treatment, respectively) (Figure 1A). Therefore, the mouse sex had no significant effect on outcome of *R. delemar* infection or treatment in this DKA model. 

We also analyzed tissue fungal burden of target organs in two independent experiments. The combined data for the lung and brain fungal burden are presented in Figure 1B. The pulmonary fungal burdens of the placebo-treated female and male mice were similar. While there was a trend towards reduced fungal burden in the brains of the control female mice relative to the male mice, this difference did not achieve statistical significance. Consistent with the survival data, both antifungal drugs reduced fungal burden in the lung when compared to placebo-treated mice, with L-AMB outperforming POSA (>1.0 log-fold and 0.5 log reduction in L-AMB- (*p* < 0.0001) and POSA-treated mice (*p* < 0.05 for male mice and *p* < 0.08 for female mice), respectively) (Figure 1B). While L-AMB also reduced the brain fungal burden by >1.0 log-fold (*p* < 0.0001) in both sexes, POSA caused minimal reduction of ~0.4 log-fold in female mice (*p* < 0.03) but not male mice (*p* = 0.14) when compared to their respective placebo groups. 

For immunological responses, due to the immunosuppression status of the mice, cytokine analysis of whole serum was not measurable with the overwhelming samples registering values below the limit of detection (LOD ranged from 0.45–134 pg/mL based on the cytokine being measured). However, cytokine analysis of lung homogenates showed measurable levels of TNF-α, IL-23p19, IL-17A, IL-12p70, IL-6, IL-10, IL-1B, IL-1α, IFN-γ, GM-CSF, CXL1, CXCL-2, CXCL10/CRG-2, CCL2, CCL3, CCL4 and CCL5 (Appendix A). In general, there were no remarkable differences in cytokine levels between female and male mice infected/untreated or those treated with L-AMB or POSA. However, there was a 4-fold increase in interferon-γ among POSA-treated female mice vs. POSA-treated male mice (*p* = 0.01) and a very small drop of ~25% in the levels of IL-12p70 among female placebo mice vs. male mice (*p* = 0.02). 

### 3.2. DKA Mice Infected with M. circinelloides

In two independent experiments with similar results, female DKA mice were more resistant to *M. circinelloides* infection when compared to male DKA mice (median survival time of 6 days vs. 3 days and a 100% mortality on day 12 vs. day 9 for female and male mice, respectively, *p* < 0.0001) (Figure 2A). Similar to what we found in DKA mice infected with *R. delemar*, both L-AMB and POSA prolonged overall survival of mice infected with *M. circinelloides*, with L-AMB outperforming POSA. Specifically, L-AMB-treated mice had an overall survival of ~80% vs. ~40% of POSA-treated mice (*p* < 0.02 of L-AMB-treated vs. POSA-treated and regardless of the mouse sex). Unlike placebo-treated mice, no differences were seen in the survival of female and male DKA mice when treated with either L-AMB or POSA (*p* = 0.695 for L-AMB-treated female vs. male mice, and *p* = 0.392 for POSA-treated female vs. male mice) (Figure 2A). 

The tissue fungal burden experiments showed that sex had no detectable effect on the lung and brain fungal burden in mice infected with *M. circinelloides* (Figure 2B). Additionally, in two independent experiments with DKA mice infected with *M. circinelloides*, both POSA and L-AMB reduced lung and brain fungal burden vs. placebo treated mice regardless of the mouse sex. Specifically, POSA-treated female mice had ~0.5-log reduction in lung and brain fungal burden vs. placebo (*p* = 0.02), a trend in reduced fungal burden was observed with POSA-treated male mice vs. placebo but that reduction did not reach statistical significance (*p* ≤ 0.09) (Figure 2B). Concordant with the survival studies, L-AMB outperformed POSA in these studies. L-AMB-treated female or male DKA mice had ~1.0 to 2.0-log reduction in lung and brain fungal burden vs. placebo (*p* < 0.001) (Figure 2B). However, unlike the difference seen in survival between placebo female and placebo male mice, no major differences in tissue fungal burden among the organs tested were noticed between the two sexes and regardless of the treatment applied (*p* > 0.05 between female and male mice) (Figure 2B).

To analyze the effects of sex on the immunological response to infection, we first attempted to use Luminex technology to measure the levels of cytokine analysis of whole serum. However, all serum cytokine levels were below the limits of detection. Next, we measured the cytokine levels in lung homogenates and detected measurable levels of TNF-α, IL-23p19, IL-17A, IL-12p70, IL-6, IL-10, IL-1B, IL-1α, IFN-γ, GM-CSF, CXL1, CXCL-2, CXCL10/CRG-2, CCL2, CCL3, CCL4 and CCL5. Infection of female mice with *M. circinelloides* but not males resulted in increased levels of CCL2, CCL3 and CCL4 (Appendix A). These chemokines are mainly associated with recruitment of phagocytes [22,23] and might explain the relative resistance of female mice to mucormycosis when compared to male mice. Notably, infection of both male or female mice with *M. circinelloides* resulted in reduced levels of CCL5, a chemokine usually associated with recruitment with T-cells, eosinophils and basophils [24,25,26]. None of the other tested cytokines showed any enhanced levels over uninfected mice.

### 3.3. Neutropenic Mice Infected with M. circinelloides

Because a slight difference was observed in female and male DKA mice infected with *M. circinelloides*, we investigated if sex differences in susceptibility to mucormycosis were also present in the neutropenic mouse model infected with this fungus. We conducted two independent experiments and present the combined data because the results of each experiment were similar. Unlike the results obtained with the DKA mouse model, we did not observe any significant differences in the survival of female or male neutropenic mice infected with *M. circinelloides* and treated with placebo, L-AMB or POSA (Figure 3A). Consistent with our previously reported data using this model [9], both antifungal agents demonstrated enhanced overall survival vs. placebo regardless of mouse gender (60%–80% survival for L-AMB-treated or 30%–40% survival for POSA-treated vs. 0% survival for placebo; *p* < 0.002 for either drug vs. placebo) (Figure 3A). Furthermore, L-AMB consistently resulted in 1.5–2.0-log reduction of lung and brain fungal burden in both female and male mice vs. placebo (*p* < 0.0001). POSA also reduced fungal burden (~1.0-log) in lung, but not the brain, of infected mice when compared to placebo (*p* < 0.01), but in general this effect was inferior to LAMB (Figure 3B). These results confirm the reported superiority of L-AMB over POSA in treating mucormycosis. Importantly, and consistent with the survival studies, no gender differences were noticed in these studies. 

Cytokine analysis of lung homogenates, but not sera, showed measurable levels of TNF-α, IL-23p19, IL-17α, IL-1β, IL-1α, IFN-γ, CXCL1, CCL2, CCL3, CCL4 and VEGF. Subtly elevated levels of CCL2 were detected in placebo-male compared to placebo-female mice. Consistently lower levels of inflammatory cytokines including IL-17α, IFN-γ and IL-1α were detected in POSA-treated female mice vs. POSA-treated male (Appendix A). However, these differences in cytokines levels are not of large magnitude and they appear to have little effect, if any, on the course of infection as evident by the similar survival and tissue fungal burden among female and male neutropenic mice.

## 4. Discussion

In medical mycology, few sex-related differences have been reported. One example where differences clearly exist between males and females is in infections caused by *Paracoccidioides* species, which are endemic in certain areas of South America [27,28,29]. The severity of incidence and severity of disease caused by this fungal species are both greater in males than in females, both in patients and experimentally infected animals [27,28,29,30]. Studies have demonstrated that this difference between the sexes is related to both the endocrine and immune responses. Estrogen has a direct effect on the growth of the fungus, as well as inhibiting the conversion to the yeast morphology. The immune response is also considered to be sexually dimorphic, as female develop more effective humoral and cell-mediated pro-inflammatory responses than males [31,32,33,34]. For example, in the asymptomatic form of paracoccidioidomycosis, a T-helper type 1 (Th1) response occurs, while a T-helper type 2 (Th2) response has been associated with severe disease. Interestingly, the endocrine and immune responses may be linked, as estrogen has been shown to stimulate pro-inflammatory cytokines (e.g., IL-12, IFN-γ and TNF-α), and down-regulate IL-10 [30]. In contrast, the synthesis of IL-10 may be enhanced by testosterone. 

While the Th1/Th2 paradigm has not been extensively studied in relation to mucormycosis, studies conducted on healthy individuals have shown the presence of specific anti-*Rhizopus oryzae* (*R. arrhizus*) T-cells that are characterized as memory Th1 cells that are capable of producing interferon-γ and enhance the anti-Mucorales phagocyte activity [35]. This study indicates that similar to the other mold infection, aspergillosis [36,37,38,39], a Th1 immune response is likely to be protective against mucormycosis. Therefore, females might be more resistant to mucormycosis than males. This assumption is modestly supported by two studies. The first is a review of 929 reported cases of mucormycosis that have been reported in the English-language literature between 1885 and 2005 which showed that mucormycosis in males represented 65% of the reported cases vs. 35% in females [7]. Additionally, a recent study that evaluated the newly approved antifungal agent isavuconazole in treating mucormycosis, showed that 81% of the enrollee were males, indicating that the disease is potentially more prevalent in males than females [40]. 

We used three models of intratracheal mucomrycosis infection to study the effect of mouse sex on infection and antifungal treatment outcome, and on host immune response. These included DKA mice infected with either *R. delemar* or *M. circinelloides*, and neutropenic mice infected with *M. circinelloides.* These two isolates are among the most commonly isolated Mucorales from patients [41]. The only evidence for an effect of mouse gender on the infection outcome was noticed with DKA female mice being more resistant than male mice when infected with *M. circinelloides* (Figure 2A). However, this difference in susceptibility to infection was very subtle and did not appear to have an effect on the outcome of antifungal treatment. Additionally, this subtle resistance of female DKA mice to infection was not detected when *R. delemar* was used to infect the DKA mice. Consistent with little effect of mouse sex on the infection/treatment outcome, no differences between female and male mice were noted in neutropenic mice infected with *M. circinelloides*. Furthermore, few differences were detected in cytokine levels among both sexes in the DKA or neutropenic mice. However, it appeared that in the DKA mouse model, the infection induced more inflammatory cytokines and chemokines involved in neutrophil (CXCL1) [23] and macrophage (CXCL2) [42] recruitment vs. uninfected mice and regardless of the mouse gender. As expected, this enhanced inflammatory response was reduced in the neutropenic mouse model. Thus, there is little effect, if any, of mouse gender on the outcome of murine mucormycosis using these two models. Based on these results, it is possible to indifferently use male, female mice or a combination of both for animal models of invasive mucormycosis when using the outbred ICR mice. Outbred mice are preferred to inbred mice in treatment experiments to account for inter-individual variability. 

Our studies with the DKA and neutropenic mouse models also confirmed previously reported data of the superiority of L-AMB when compared to POSA treatment [9,43]. These results also reflect the recent Global guidelines of using L-AMB as a first-line therapy and reserving POSA treatment for salvage therapy [6]. It is also reassuring that the female and male mice responded to L-AMB or POSA treatment favorably in both models and with two different agents of mucormycosis.

## Figures and Tables

**Figure 1 jof-07-00313-f001:**
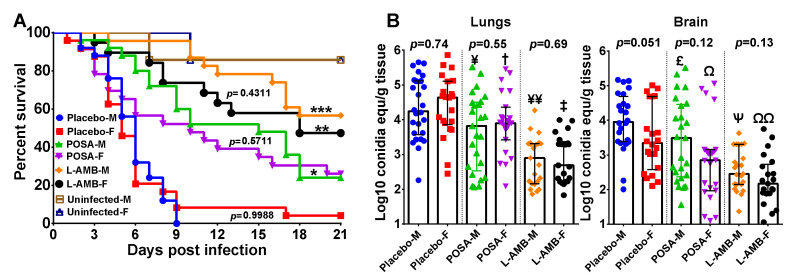
Female and male DKA mice are equally susceptible to *R. delemar* infection. (**A**) Survival of female (F) or male (M) DKA mice combined from two independent experiments infected with *R. delemar* 99-880. Mice (*n* = 21 to 25 per group, except the uninfected control which had 7 for each sex) were made diabetic and infected. * *p =* 0.0003 for POSA-M vs. Placebo-M. ** *p =* 0.0008 for L-AMB-F vs. Placebo-F. *** *p* < 0.05 for L-AMB-M vs. Placebo-M or POSA-M. *p* values on the figure compare female and male mice within the same treatment. (**B**) Combined tissue fungal burden (qPCR) in lung and brain of mice infected with *R. delemar.* Mice (*n* = 20 per group from 2 experiments) were infected, treated and organs harvested at Day 4 post infection. Lung CFU: ^¥^
*p* = 0.03 for Placebo-M vs. POSA-M, ^†^
*p* = 0.08 for Placebo-F vs. POSA-F, ^¥¥^
*p* < 0.006 for L-AMB-M vs. Placebo-M or POSA-M, ^‡^
*p <* 0.0001 for L-AMB-F vs. Placebo-F or POSA-F. Brain CFU: ^£^
*p =* 0.135 for POSA-M vs. Placebo-M, ^Ω^
*p =* 0.02 for POSA-F vs. Placebo-F, ^Ψ^
*p* = 0.01 for L-AMB-M vs. Placebo-M or POSA-M, ^ΩΩ^
*p* < 0.0001 for L-AMB-F vs. Placebo-F and *p* = 0.1 for L-AMB-F vs. POSA-F.

**Figure 2 jof-07-00313-f002:**
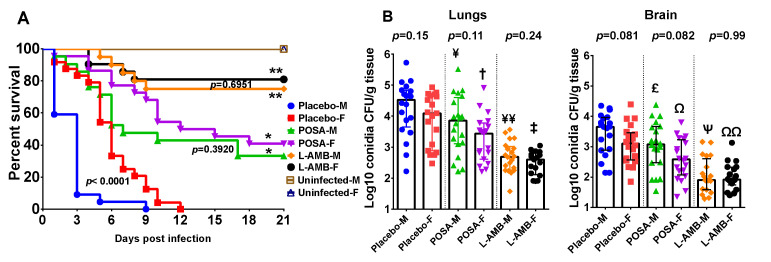
Female DKA mice are slightly more resistant to *M. circinelloides* infection than male DKA mice. (**A**) Survival of female (F) or male (M) DKA mice combined from two independent experiments infected with *M. circinelloides* f. *jenssenii* DI15-131. Mice (*n* = 22–24 per group, except the uninfected control which had 10 for each sex) were made diabetic and infected. * *p* < 0.03 for POSA-F or POSA-M vs. Placebo of the corresponding sex. ** *p* < 0.02 for L-AMB-F or L-AMB-M vs. Placebo or POSA of the corresponding sex. *p* values on the figure compare female and male mice within the same treatment. (**B**) Combined tissue fungal burden (qPCR) in lung and brain of mice infected with *M. circinelloides*. Mice (*n* = 20 per group from 2 experiments) were infected, treated and organs harvested at Day 4 post infection. Lung CFU: ¥ *p* = 0.08 for POSA-M vs. Placebo-M, † *p* = 0.02 for POSA-F vs. Placebo-F, ¥¥ *p* < 0.0003 for L-AMB-M vs. Placebo-M or POSA-M, ‡ *p* < 0.001 for L-AMB-F vs. Placebo-F or POSA-F. Brain CFU: ^£^
*p* = 0.1 for POSA-M vs. Placebo-M, Ω *p* = 0.02 for POSA-F vs. Placebo-F, Ψ *p* < 0.0001 for L-AMB-M vs. Placebo-M or POSA-M, ΩΩ *p* < 0.005 for L-AMB-F vs. Placebo-F or POSA-F.

**Figure 3 jof-07-00313-f003:**
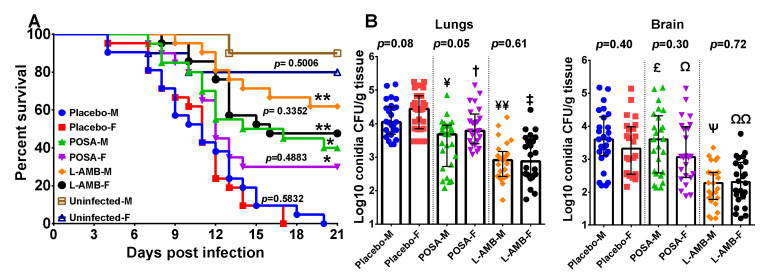
Female and male neutropenic mice are equally susceptible to *M. circinelloides* infection. (**A**) Survival of female (F) or male (M) neutropenic mice combined from two independent experiments infected with *M. circinelloides* f. *jenssenii* DI15-131. Mice (*n* = 21 per group, except the uninfected control which had 10 for each sex) were made neutropenic and infected. * *p* = 0.002 for POSA F or POSA M vs. Placebo of the corresponding sex. ** *p* <0.001 for L-AMB-F or L-AMB-M vs. Placebo of the corresponding sex. *p* values on the figure compare female and male mice within the same treatment. (**B**) Combined tissue fungal burden (qPCR) in lung and brain of mice infected with *M. circinelloides*. Mice (*n* = 26 in each group from 2 experiments) were infected, treated and organs harvested at Day 4 post infection. Lung CFU: ^¥^
*p* = 0.001 for POSA-M vs. Placebo-M, ^†^
*p* = 0.005 for POSA-F vs. Placebo-F, ^¥¥^
*p* < 0.01 for L-AMB-M vs. Placebo-M or POSA-M, ^‡^
*p* < 0.0001 for L-AMB-F vs. Placebo-F or POSA-F. Brain CFU: ^£^
*p* = 0.75 for POSA-M vs. Placebo-M, ^Ω^
*p* = 0.53 for POSA-F vs. Placebo-F, ^Ψ^
*p* < 0.0001 for L-AMB-M vs. Placebo-M or POSA-M, ^ΩΩ^
*p* < 0.005 for L-AMB-F vs. Placebo-F or POSA-F.

## Data Availability

The data presented in this study are available in this article and the Appendix A. Source data are available from the corresponding author upon request.

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
