# Peer review of "Evaluation of Sex Differences in Murine Diabetic Ketoacidosis and Neutropenic Models of Invasive Mucormycosis"

_jof, 2021, doi:10.3390/jof7040313_

Round 1

Reviewer 1 Report

Dear authors,

Thank you very much for the interesting paper “Evaluation of Sex Differences in Murine Diabetic Ketoacidosis 2 and Neutropenic Models of Invasive Mucormycosis”. I have the minor remarks listed below.

Introduction

Line 47: Please explain to the non-US reader what FY2021 is.

Materials and Methods

Please precise locations of bought animals or drugs. For example line 90, Gilead; line 95, Evigo, line 107, Sigma..

Line 99: Did you mean “filter sterilized”?

Line 100: Please delete round brackets from (210 mg/kg).

Line 126-129: Why did you do that?

Line 130-131: In this sentence something seems to be missing.

Line 130: To my opinion at this point has to be added how many animals were used in the different groups (same as in figures). Additionally it has to be explained why different numbers of animals were used for the groups.

Please add how often the experiments were performed.

Results

Line 156: Please write “both” with a lowercase letter.

In the figure legends of figure 2 and 3 M. circinelloides is not always written in italic. Please correct this.

Line 236: For me it is not clear why reference 27 is cited here.

Author Response

Reviewer 1

Thank you very much for the interesting paper “Evaluation of Sex Differences in Murine Diabetic Ketoacidosis 2 and Neutropenic Models of Invasive Mucormycosis”. I have the minor remarks listed below.

We thank the reviewer for the favorable comments. 

Introduction

Line 47: Please explain to the non-US reader what FY2021 is.

This is now spelled out in the revised version. 

Materials and Methods

Please precise locations of bought animals or drugs. For example line 90, Gilead; line 95, Evigo, line 107, Sigma..

We have added the precise locations as requested by the reviewer

Line 99: Did you mean “filter sterilized”?

Yes and this is now corrected.

Line 100: Please delete round brackets from (210 mg/kg).

Done

Line 126-129: Why did you do that?

This is done to enumerate the delivered fungal spores to the lungs. When animals are sacrificed immediately after infection, fungal cells are still in its spore formation and have not germinated. This enables enumeration by traditional homogenization followed by culturing.   We added the following statement to the start of the sentence “To determine the delivered fungal inoculum to the lungs”

Line 130-131: In this sentence something seems to be missing.

Addition of “To determine the delivered fungal inoculum to the lungs” now explains the sentence better.

Line 130: To my opinion at this point has to be added how many animals were used in the different groups (same as in figures). Additionally it has to be explained why different numbers of animals were used for the groups.

The number of animals in the groups is reliant on how many mice respond to the immunosuppression and this number is not always an even number.  For example, only ~80-85% of the mice become diabetic after streptozotocin injection.

As for the suggestion of the reviewer, we now mention that at least 10 mice per group were used each experiment. Lines 150-152 of the revised version.

Please add how often the experiments were performed.

All experiments were repeated once.  This is now mentioned in the revised manuscript (Lines 150-152).

Results

Line 156: Please write “both” with a lowercase letter.

Done

In the figure legends of figure 2 and 3 M. circinelloides is not always written in italic. Please correct this.

This is now corrected.

Line 236: For me it is not clear why reference 27 is cited here

This was an unintentional mistake.  The error is corrected and the references are reformatted.

Reviewer 2 Report

General comments

In this paper the authors have evaluated the sex differences in two murine models of mucormycosis. The aim of the study is very interesting. They used three different models: Rhizopus and Mucor infections in diabetic mice and Mucor infection in neutropenic mice.

Overall, it is shown that there are no or marginal differences between sexes.

the paper is clearly presented. In summary, the study has been well conducted and that provide useful data.

Specific comments

  1. Line 67: Could you comment on the reasons that most of the previous studies used male mice.
  2. Lines 149, 181, 224 Please detail what similar means: were the survival curves from the two experiments not statistically different? This is important to allow the pool of data obtained from independent experiments.
  3. Line 172: Please state here why serum levels were not measurable. Was this linked to technical problems or to blood levels under the limit of detection? Please provide, at least, a rough estimate of the limit of detection.
  4. Discussion and conclusions: Based on their results, the authors could propose practical recommendations such as: -It is possible to indifferently use male or female mice for animal models of invasive mycosis. In treatment experiments, it is generally recommended to use outbred mice to account for inter-individual variability and to strengthen results. Would it be reasonable to use male and female in the same experiment? Even though there are subtle differences between the sexes, can this variability be beneficial and enhance the strength of an experiment's results?
  5. Line 170: typo: “female male”
